# Bioelectrochemical Purification of Biomass Polymer Derived Furfural Wastewater and Its Electric Energy Recovery

**DOI:** 10.3390/polym15163422

**Published:** 2023-08-16

**Authors:** Hailing Tian, Yue Quan, Zhenhao Yin, Chengri Yin, Yu Fu

**Affiliations:** 1Department of Chemistry, Yanbian University, Yanji 133002, China; 2Laboratory Animal Center, Yanbian University, Yanji 133002, China; 3Department of Environmental Science, Yanbian University, Yanji 133002, China

**Keywords:** biomass polymer conversion, electric energy recovery, furfural wastewater, microbial fuel cell

## Abstract

With the increasing environmental pollution caused by waste polymers, the conversion of polymer components in biomass into valuable products is of great significance for waste management and resource recovery. A two-stage microbial fuel cell (MFC) was used to treat furfural wastewater in this study. The maximum output voltage was 240–250 mV and the power generation time in an operation cycle was 286 h. The degradation efficiency of furfural reached 99–100% (furfural concentration at 300–3000 mg/L) and was slightly reduced to 91% at 7000 mg/L. In addition, the BOD/COD ratio of the furfural wastewater increased from 0.31 to 0.48 after MFC processing. The molecular analysis of the anodic bacterial isolates indicated that the phylogenetic bacterial mixture was dominated by five active anaerobic bacteria with a similarity percentage above 99% for each strain: *Burkholderia* (*B. burdella*), *Clostridium sensu stricto* (*Cymbidaceae*), *Klebsiella* (*Klebsiella*), *Ethanoligenens* (anaerobic genus), and *Acidocella* (anaerobic genus); the mixture exhibited good properties to carry out bioelectricity generation in the microbial fuel cell. This indicates that the MFC has effectively degraded furfural for pollutant removal and power generation and is a promising clean method to treat furfural pollution in industry wastewater.

## 1. Introduction

Water pollution caused by global population growth and industrial expansion has become a serious problem hindering fresh water supplies. Polymers from biomass waste including plant/forest waste, biological industrial process waste, municipal solid waste, algae, and livestock waste are potential sources for renewable and sustainable resources [1]. Lignocellulose is the most abundant biomass, mainly comprising cellulose, hemicellulose, lignin, and other polymers. Furfural mainly comes from hemicellulose that is rich in natural polymers, and can be produced on a large scale with low-cost corn cobs, straw, peanut shells, and other agricultural by-products as raw materials. Furfural is an important organic compound directly/indirectly utilized to synthesize more than 1000 chemical products, and is widely used in the petroleum, chemical, pharmaceutical, food, and other industries as an important organic chemical raw material intermediate and chemical solvent [2]. The global production of furfural is estimated to be 2.8 million tons per year, of which 90% is from China, the Dominican Republic, and South Africa. The total production capacity of furfural in China exceeds 600,000 tons, accounting for about 80% of the global total production capacity. Although widely synthesizing furfural significantly improved the added value of waste biomass polymers, the wastewater generated from its synthesis process may cause serious environment and sanitation issues. An investigation indicated that 1 ton of furfural production can lead to about 20–30 tons of furfural wastewater generation, and there are over 200 factories in China which will lead to approximately 4–6 million tons of furfural wastewater generation annually [3]. In addition, furfural wastewater is a highly acidic (contains sulfuric acid and acetic acid) organic wastewater with complex components including solid residues consisting of lignin and cellulose substances, and various volatile organic compounds. Furfural wastewater discharged into the environment from different sources can cause ecotoxicity [4]. When humans are exposed to furfural (over 8 mg/m^3^), it can lead to substantial long-term health consequences, including eye irritation, nausea, liver and kidney disorders, headaches, weakness, central nervous system disorders, mutations, and tumors [5,6]. Therefore, the sustainable treatment of furfural wastewater as well as recovering the chemical energy potential inside its substrates is of great importance for both environmental and resource concerns.

Various methods, including physical, chemical, and biological methods have been used for furfural wastewater treatment [7,8,9,10,11,12,13,14,15]. However, the conventional physical and chemical methods are usually limited by a high energy demand, ineffectiveness of catalysis transmission, the potential risk of generating toxic by-products and environmental pollution, as well as the high cost. The development of new methods for the efficient processing of furfural wastewater has become a pressing issue. Recently, biological processes are considered a promising approach because they have flexible and reliable performance, have relatively high economic benefits, use environmentally friendly technologies, are able to degrade contaminants to less toxic or harmful materials, etc. As a typical representative biological method, bioelectrochemical systems can drive the electrons generated from the biodegradation of furfural wastewater to the acceptor to recover the chemical energy from the components of the wastewater as electric energy. For example, a microbial fuel cell (MFC) could degrade pollutants and convert the chemical energy contained in pollutants into clean energy by using active microorganisms as a biocatalyst in an anaerobic anode compartment for the production of bioelectricity. Currently, MFCs are widely applied for treatment of wastewaters, such as oil [16], phenol/acetone [17], textile [18], municipal [19], and saline wastewaters [20].

To the best of our knowledge, there are few reports on the application of MFCs in furfural wastewater treatment. Therefore, in the present study, we provided a case study of using a self-made, two-chamber, cube-type, lab-scale MFC for furfural wastewater purification and energy recovery, by effectively degrading the biomass-polymer-derived furfural wastewater and using it for electric energy production.

## 2. Materials and Methods

### 2.1. Reactor Design and Construction

The self-made MFC reactor was made of Perspex frames by Beijing Hon Haitian Technology Co., Ltd. (Beijing, China), which consist of an anode chamber and a cathode chamber with an effective volume of 252 mL (Figure 1). The anode and cathode compartment were separated by a proton exchange membrane (PEM, Nafion 212, Dupont Co., Wilmington, DE, USA). The anode electrode was made of carbonbrush (3 cm × 5 cm, Hubei Jingzhou Hot New Materials Co., Zhengzhou, China), while the cathode electrode was made of carbon felt (5 mm thick, Beijing Honghaitian Technology Co., Beijing, China). Copper wires were used to connect the circuit with an external resistance and all wire contacts were sealed with an epoxy material. A peristaltic pump (YZ15, Chongqing Knap Co., Chongqing, China) was connected to the inlet/sampling hole above the anode chamber, while the cathode chamber was provided with an aerator that was externally connected to an aeration pump (SB-988, Beijing Honghaitian Technology Co., Beijing, China). All materials were sterilized with medicinal alcohol, and the MFC was operated at 30 °C. During the start-up periods, activated sludge superannuate was added into the MFC anode chamber with an enrichment medium.

### 2.2. Inoculation and MFC Operation

The anode chambers of MFC were inoculated with the activated sludge supernatant mixed with a nutrient solution from a sewage treatment plant in Yanji, China. The enrichment medium was placed in the anode chamber of the assembled MFC reactor, and the pre-prepared carbon-free buffer (PSB) solution was added to the cathode chamber. After 30 days (the inoculation period), the anodic biofilm was comprised of electricity-producing microbes. The enrichment medium composition (per liter) was: 1.00 g glucose, 0.46 g Na_2_HPO_4_, 0.25 g NaH_2_PO_4_, 0.03 g NH_4_Cl, 0.01 g KCl, 12.5 mL trace solution, and 500 mL sludge supernatant. The trace solution consisted of (per liter): 0.05 g MgSO_4_·7H_2_O, 0.01 g NaMoO4, 0.01 g CuSO_4_·5H_2_O, 0.03 g KI, 0.01 g NiCl_2_·6H_2_O, 0.01 g H_3_BO_3_, 0.2 g ZnSO_4_·7H_2_O, 0.1 g CoCl·6H_2_O, 1.5 g FeCl·6H_2_O, and 0.01 g NaSeO_6_. After connecting and electrifying with the paperless recorder and resistance box, the whole MFC was put into the thermostatic water bath and the temperature was adjusted to 30 °C. The real-time voltage was recorded every 20 s by a paperless recorder. Power generation performance of the MFC was tested by using 1 g/L of glucose as the substrate (pH 7.0). The external resistance (R) of the MFC was set at 500 Ω to continuously record the generated voltage (U) with a recording interval of 20 s and output the data. The fuel cell was immersed in a thermostatically controlled water bath at 30 °C.

### 2.3. Composition of Biomass Polymer Derived Furfural Wastewater

Organic matter content is one of the legally regulated parameters in wastewater treatment processes. Chemical and biochemical oxygen demand (COD and BOD, respectively) are the main parameters used to determine wastewater organic matter content. The oxidizability of samples (i.e., COD and BOD) was determined according to the National Standards as described previously [21,22,23]. The dissolved oxygen (DO) was measured with an Oxi-X meter (WTW Company, Germany). The total nitrogen (TN) and total phosphorus (TP) were measured according to the standard methods [21,22,23]. The lignocellulosic (hemicellulose, cellulose, and lignin) components were determined by ANKOM A200i Semi-automatic fiber analyzers and analyzed by the Van Soest method [24]. For this, the furfural wastewater samples provided by a company in the city of Jilin (China) was placed in a 1.0 L cylinder and allowed to settled for 30 min. The 100 mL supernatant of furfural wastewater sample was centrifuged and the supernatant was usedto measure common chemical parameters including TN, TP, COD, and BOD. Temperature, pH, DO, and conductivity of the furfural wastewater sample were measured in situ. The data on the pollutant concentration were recorded for 50.0 mL samples withdrawn over time starting at the beginning of operation of the MFC. All analyses were performed in triplicate.

### 2.4. Analytical Methods

The organic components present in the furfural wastewater samples and furfural wastewater treatment samples collected from MFC were analyzed using GC and GC-MS (Thermo-Finnigan SSQ710, San Jose, CA, USA) in a semi-quantitative manner without internal or external standards. Xcalibur software was used for substance identification in the GC-MS data analysis. The standard curve of furfural was analyzed by an HP-1100 high-performance liquid chromatography system (Agilent Technologies, Santa Clara, CA, USA) with an ultraviolet detector. The surface area of the anode electrode was measured using a UT52 multimeter (Uni-Trend Technology Ltd, Dongguan, Guangdong, China) and the area current density (IA) and power density (PA) were calculated. The polarization curve and the power density curve were plotted by changing the external resistance. The power density (PD) and polarization curve of the MFC were plotted by changing the external resistance (50–9999 Ω) following a reported method [17]. The electrochemical performance of the MFC was evaluated through the expression V = IR (Ohm’s Law), where V is voltage (V), I is current (A), and R is external resistance (Ω). Power (P) in watts (W) was calculated from P = IV. The power density was calculated according to the electrodes’ total surface area. The internal resistance was determined from the slope of the polarization curve. All analyses were carried out in triplicate and the final result was the average of the tested values.

### 2.5. Microbial Community Analysis

A 1.0 mL volume of the anolytic bacterial mixed culture was sampled from the anode zone and extracted using a PowerSoil^®^ DNA Isolation Kit (MoBio Laboratories Inc., Carlsbad, CA, USA) under aseptic conditions. The bacterial ribosomal (16S rRNA) gene was identified according to its sequence by using the universal primers 1492R (5′-CGG TTA CCT TGT TAC GAC TT-3′) and 27F (5′-AGA GTT TGA TCM TGG CTC AG-3′), following previous studies [25,26,27]. The PCR products were sequenced by Sangon Biological Engineering Co. (Shanghai, China). A metagenomics method was utilized to unravel the microbial community contained in the sludge, and used the polymerase chain reaction followed by high-throughput sequencing (Hiseq 2500, Illumina, Inc., San Diego, CA, USA).

### 2.6. Statistical Analysis

All experiments were conducted in triplicate. The results are reported as the mean and ± tandard deviation and were subjected to basic statistical analysis of variance (one-wayANOVA) using the commercial statistical software Origin Pro 9.0. A differencewas considered significant if *p* < 0.05 with a 95% confidence interval [28].

## 3. Results

### 3.1. Start-Up Procedure

Activated sludge superannuate was added into the MFC anode chamber with an enrichment medium. The reactors were first conducted under a batch-fed mode (10 d) with an external resistance of 1000 Ω to inoculate the microorganisms in both chambers. At the beginning of this stage, almost no output voltage was detected. After 167.5 h (7 days) of continuous incubation, the voltage rose to 60 mV. There was no need to add bacteria and only the glucose culture solution was changed when the voltage was lower than 50.0 mV. The output voltage reached a maximum value of 257.1 mV at 698 h (Figure 2) and when the background value of 52 mV was exceeded (after 10 days), the enriched electrogenic bacteria can be considered positive [29]. Therefore, when the running time reached about 750 h (about 31 days), the enrichment was stopped.

### 3.2. Electricity Generation

The electrical properties of the MFC were evaluated using 7000 mg/L furfural wastewater as a matrix and the related parameters are shown in Figure 3. According to linear fitting of the voltage data, the linear equation is y = −2598x + 118.1, R^2^ = 0.995 (x represents the current; y represents the voltage). The current density and power density data were calculated to plot the power density curve. The power density first increased and then decreased with increasing current density. The maximum power density was 0.132 mW/m^2^, and the slope of the fitted line is equal to the maximum power corresponding to a resistance of about 2000 Ω. It can be deduced from the results that the internal resistance of the MFC was about 2300 Ω.

The internal resistance and the PD was relatively low in our study. This may be due to the larger surface area of the carbon brush that is relatively rich in microorganisms. This also results in the dense growth of the internal microbial community, hindering the renewal of the microbial community and resulting in reduced mass transfer efficiency, more microorganisms, and a relatively large internal resistance, which leads to a low overall PD.

### 3.3. Biomass-Polymer-Derived Furfural Wastewater Treatment Efficiency

#### 3.3.1. Characterization of Biomass-Polymer-Derived Furfural Wastewater

Currently, furfural is prepared by the acid-catalyzed dehydration of pentose sugars from the polymeric components in lignocellulosic biomass (e.g., corn stalk and bagasse) under high temperatures. Furfural wastewater is a highly acidic (contains sulfuric acid and acetic acid) organic wastewater with complex components including solid residues consisting of lignin and cellulose, and various volatile organic compounds. In this research, the data on water quality parameters of the furfural wastewater and in the MFC are given in Table 1. After MFC treatment, the content of cellulose, hemicellulose, lignin, sulfuric acid, acetic acid, and furfural in the biopolymers decreased significantly. The BOD/COD ratio of furfural wastewater also increased from 0.31 ± 0.01 to 0.48 ± 0.02 after MFC treatments due to the removal of sulfide and cyanide.

#### 3.3.2. Standard Curve of Furfural

The standard curve was prepared by performing a dilution series of furfural concentrations (4000, 3000, 2000, 1000, 500, 250, 100, 50, 25, 10, 5, and 1 mg/L). In the relationship between the furfural concentration and peak area, the linear correlation coefficient was high. Calculated from the equation, the linear equation is y = 15,404x + 1,655,330, R^2^ = 0.988 (x represents the furfural concentration; y represents the peak area) (Figure 4).

#### 3.3.3. Removal Performance of Biomass-Polymer-Derived Furfural Wastewater

The MFC was started by adding glucose to simulated furfural wastewater with different concentrations of a furfural solution and the voltage was recorded (Figure 5). The maximum output voltage was maintained around 240–250 mV. The higher the concentration of furfural, the longer the maximum voltage lasted. The time when the voltage over 100 mV increased with the increase in furfural concentration. This trend was more obvious at a furfural concentration of 1000 mg/L (C) to 7000 mg/L (F). When the concentration of furfural was 300 mg/L (A), the electricity production time was the shortest (12 h). When the concentration of furfural was 1000 mg/L (B and C), the electricity generation period for a 100 mV voltage was about 60 h. As the concentration increases from D to F, the electricity generation cycle also increases to 117 h, 190 h, and 286 h, respectively. The production time of furfural as a substrate reached 286 h, which is nearly 6-times longer than the previous generation using only glucose as the substrate. With the increase in furfural wastewater concentration, the time for degrading furfural increased. From stage A to E, the furfural was completely removed in each cycle and the degradation rate reached 99.9% (Figure 6). In stage F, within the first 100 h, the removal of furfural from the wastewater was 96%, and was reduced to 91% at 200 h. When the concentration of furfural was low, the electrogenic microorganisms first used the easily degradable, non-toxic glucose as the carbon source for growth, resulting in a rapid rise in voltage. These results agree with Luo et al. [30] who showed that power generation can use furfural as the substrate. The bacterial attachment and electron transfer for the energy conversion affect the removal performance of MFCs [31]. In addition, the amount of microorganisms that can degrade furfural also increased, leading to the rapid degradation of furfural, while the easy degradation of glucose also leads to a small amount of electricity production. As the concentration of glucose decreased, the concentration of furfural with high toxicity increased. When the concentration of glucose gradually decreased, the voltage of the MFC, after replacing the anode solution, rose to the maximum voltage more quickly. However, the rate at which the voltage rises to the highest voltage was significantly slower than when the glucose concentration was high, but the time relative to the power generation was significantly increased, and the voltage can be output slowly. It can be inferred that the microorganisms in the MFC can gradually adapt to the toxic environment of the furfural solution, and the dominant strain which can degrade furfural gradually increased in the replacement solution, which also indicates that glucose can accelerate the degradation of furfural to some extent. However, the voltage curve showed fluctuations with the increasing concentration of furfural after reaching the maximum and declined after a period of time. This may be due to the degradation of glucose by the electrogenic microorganisms before degrading the toxic furfural solution. This will cause voltage fluctuations. Overall, this method using an MFC can treat not only the high concentration of furfural wastewater, but the degradation effect is also very good. The degradation rate was generally not higher than 99%.

### 3.4. Microbial Morphology and Community in MFC

Five dominant bacterial genera, *Burkholderia* (*B. burdella*), *Clostridium sensu stricto* (*Cymbidaceae*), *Klebsiella* (*Klebsiella*), *Ethanoligenens* (*anaerobic genus*), and *Acidocella* (anaerobic genus), were detected in the anode chamber solution (Figure 7). Their relative abundances were 39.95%, 34.66%, 12.18%, 4.18%, and 2.33%, respectively. Among these genera, *Clostridium sensu stricto* in the anode plays important roles in carbon and nitrogen removal in coupled systems [32]. *Klebsiella* can not only reduces iron oxide and generate electricity under anoxic conditions, but it is also capable of dissimilatory nitrate reduction [33]. *Ethanoligenens* (anaerobic genus) can produce hydrogen [34], and some *Acidocella* species are known to be potential electroactive oxic microbes that can catalyze oxygen reduction [35]. The roles of the other genera need further investigation. Thus, we demonstrated that the use of a mixed bacterial culture from sludge allowed for furfural utilization in the MFC.

## 4. Discussion

Furfural is prepared by the acid-catalyzed dehydration of pentose sugars from the polymeric components in lignocellulosic biomass (e.g., corn stalk and bagasse) under high temperatures. Recently, global annual production of furfural has exceeded 750,000 tons, about 80% of which is produced in China [36]. Converting the biopolymers in waste biomass to biochar is attractive for both environmental and energy efficiency reasons [1]. MFC can convert the biopolymers in waste biomass into electricity through redox reactions catalyzed by microorganisms. The operation of MFC is affected by the materials used for the construction of the system, its structure, the microorganisms present, the substrate, and environmental conditions [37]. However, there are few reports on the application of MFCs in furfural wastewater treatment. Luo et al. [30] first reported that electricity was generated from the biodegradation of furfural in an MFC under the conditions of a low concentration of furfural (the highest concentration was 6.68 mM furfural, equivalent to 642 mg/L). The low furfural removal performance severely hindered the large-scale application of MFCs. In our study, an MFC was used to treat highly concentrated furfural wastewater (up to 7000 mg/L) and studied its electricity generation performance.

Electroactive microorganisms possess the ability to donate (electrogenic) or accept (electrotrophic) electrons from a substrate. Electrogenic microorganisms release electrons on the anode surface, being represented and quantifiable as a positive electric current; however, electrotrophic microorganisms are responsible for recovering these electrons from the cathode surface [38]. These transformation processes will vary in each species of microorganism, due to its characteristics and ease of adaptation to the environment in which they are found. This study used *Burkholderia* (*B. burdella*), *Clostridium sensu stricto* (*Cymbidaceae*), *Klebsiella* (*Klebsiella*), *Ethanoligenens* (*anaerobic genus*), and *Acidocella* (anaerobic genus). These species have been shown to have the potential to generate electricity, which could be useful for converting furfural wastewater into electricity.

The main feature of MFCs is their ability for electrical energy production. MFC reactors are mostly made at the laboratory scale and are able to produce energy ranging from 0.924 to 6492 mW/m^2^ [39]. The highly varied amounts of energy that can be harvested are strongly influenced by the type of MFC (double or single), electrode materials (anode and cathode), microbial species, and substrates used [40]. Studies using a double-chamber MFC reported a relatively high energy production (400 s/d 6492 mW/m^2^) [39,40,41], so this study also used a two-chamber MFC. The effect of substrates on electrical energy generation was also supported by two other studies that used glucose as the carbon source which not only has a quite high carbon content (1000–3000 mg/L) but is also easily degradable [40,41,42]. This study also confirmed that the use of glucose as a carbon source in MFCs could effectively promote the efficient degradation of furfuralfrom wastewater. The maximum output voltage was 240–250 mV and the power generation time in an operation cycle was 286 h. The degradation efficiency of furfural reached 99–100% (furfural concentration at 300–3000 mg/L) and was slightly reduced to 91% at 7000 mg/L. Meanwhile, the BOD/COD ratio of furfural wastewater increased from 0.31 ± 0.01 to 0.48 ± 0.02 after MFC processing. This study not only proved the efficacy of graphite electrodes, but also that the activated sludge acclimation affects the electrode’s efficiency in converting furfural wastewater into electricity. Future research on MFC should focus on achieving a higher efficiency, enhancing electrode potentials, improving the working mechanism, and identifying potential field applications. Although the power generation of furfural wastewater from MFCs was significantlyimproved in recent years, it is still only at the labscale and needs to be scaled up for commercialization for future cutting-edge applications to address the major challenges we are facing right now.

## 5. Conclusions

In this study, we investigated the bioelectrical performance of a two-chamber MFC inoculated with sludge as a microbial source and fueled with furfural wastewater as the electron donor. The results showed that MFC could effectively treat high concentration furfural wastewater, and the degradation rate was still 91% at the furfural concentration of 7000 mg/L. The maximum recorded voltage was 240–250 mV and the power generation time in an operation cycle was 286 h. The BOD/COD ratio of furfural wastewater increased from 0.31 ± 0.01 to 0.48 ± 0.01, with a removal rate as high as 90%. The obtained results were compared with the published literature and the current study demonstrated the potential of the bacterial community of furfural wastewater to produce electricity using MFCs.

## Figures and Tables

**Figure 1 polymers-15-03422-f001:**
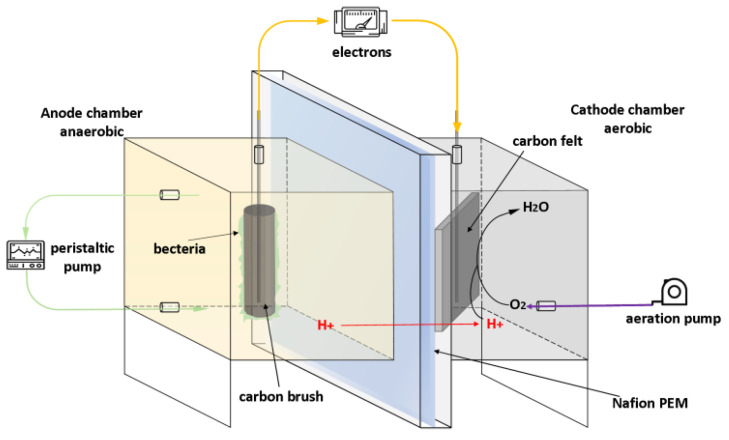
Schematic of two-chamber MFC.

**Figure 2 polymers-15-03422-f002:**
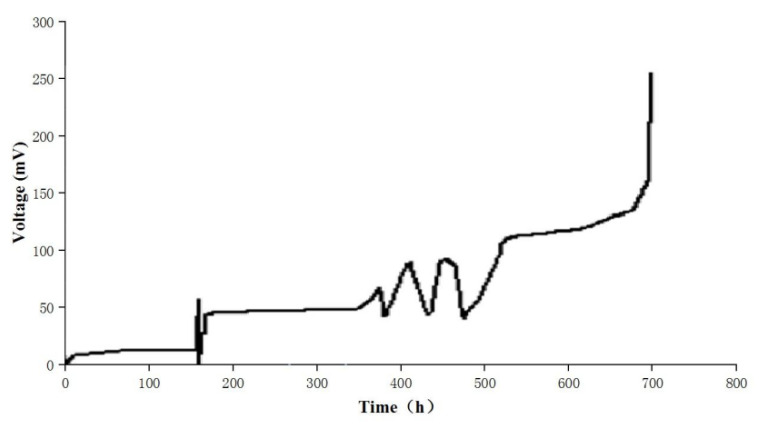
Voltage changes during the enrichment of electricity-producing microorganisms.

**Figure 3 polymers-15-03422-f003:**
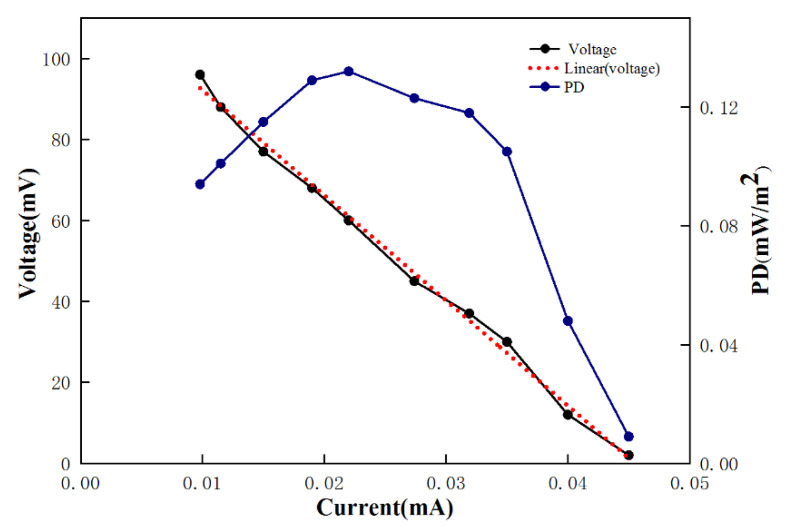
Current density versus power density (PD), polarization curve, and maximumrecorded voltages against current density for the MFC. The maximum recorded voltages and power density were found to be 96 mV and 0.132 mW/m^2^, respectively.

**Figure 4 polymers-15-03422-f004:**
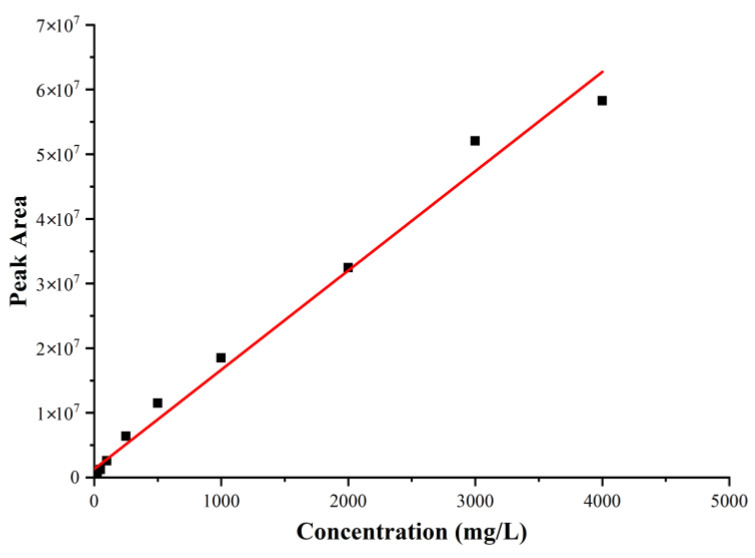
Standard curve of different furfural concentrations (4000, 3000, 2000, 1000, 500, 250, 100, 50, 25, 10, 5, and 1 mg/L).

**Figure 5 polymers-15-03422-f005:**
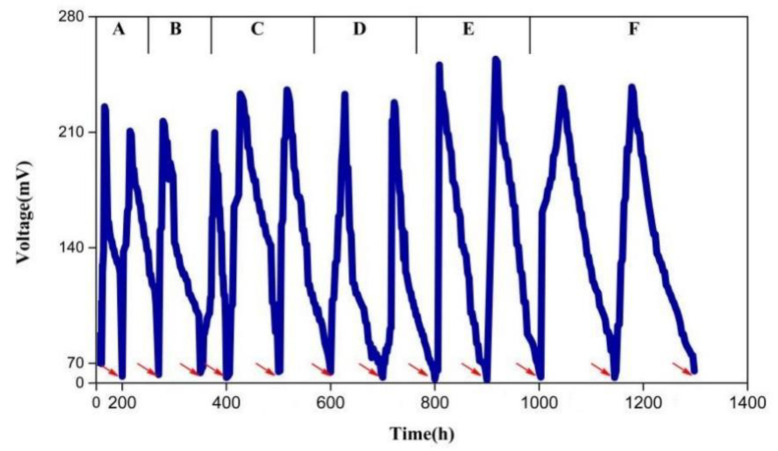
Voltage changes with different concentrations of furfural as substrate in the MFC. The red arrow represents the time when the anolyte was replaced. A: 300 mg/L furfural + 1000 mg/L sodium acetate + 1000 mg/L glucose, B: 1000 mg/L furfural + 1000 mg/L glucose, C: 1000 mg/L furfural, D: 3000 mg/L furfural, E: 5000 mg/L furfural, F: 7000 mg/L furfural.

**Figure 6 polymers-15-03422-f006:**
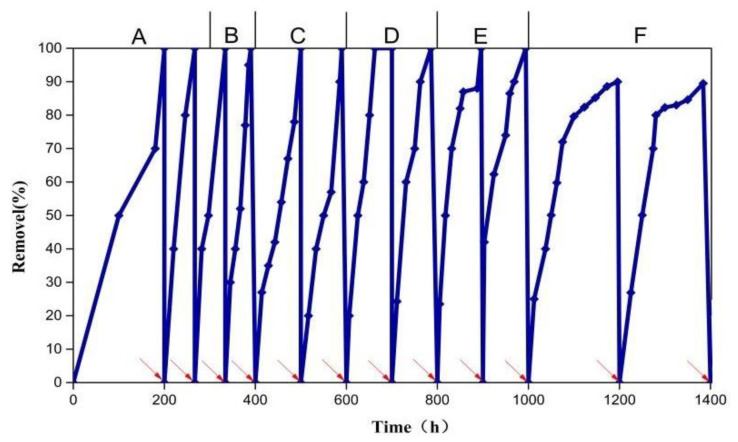
Removal of furfural from wastewater in MFC. The red arrow represents the time when the anolyte was replaced. A: 300 mg/L furfural + 1000 mg/L sodium acetate + 1000 mg/L glucose, B: 1000 mg/L furfural + 1000 mg/L glucose, C: 1000 mg/L furfural, D: 3000 mg/L furfural, E: 5000 mg/L furfural, F: 7000 mg/L furfural.

**Figure 7 polymers-15-03422-f007:**
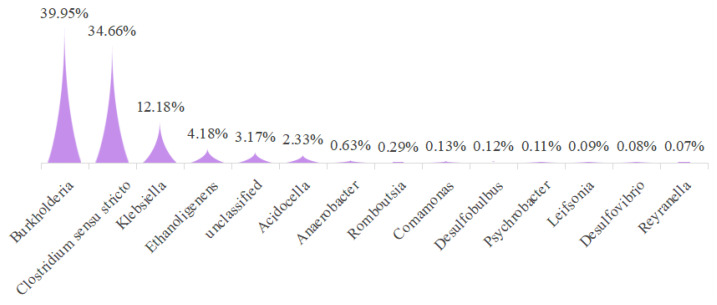
Bacterial community composition at the genus level.

**Table 1 polymers-15-03422-t001:** Key quality parameters before and afterprocessing of furfural wastewater.

Parameter (g/L)	Furfural Wastewater	After Treatment by MFC
COD	16.31 ± 0.01	1.79 ± 0.06
BOD	5.07 ± 0.01	3.73 ± 0.12
TN	1.61 ± 0.01	0.51 ± 0.05
TP	(31.90 ± 0.01) × 10^−3^	(0.16 ± 0.01) × 10^−3^
DO	(0.12 ± 0.01) × 10^−3^	(1.86 ± 0.03) × 10^−3^
pH	2.15 ± 0.02	6.10 ± 0.07
T, °C	55.00 ± 0.01	30.00 ± 2.2
H_2_SO_4_	8.27 ± 0.01	0.82 ± 0.17
CH_3_COOH	10.38 ± 0.14	1.60 ± 0.08
hemicellulose	29.73 ± 0.68	21.99 ± 0.50
cellulose	34.90 ± 0.46	30.71 ± 0.41
lignin	8.15 ± 0.22	7.50 ± 0.20
furfural	6.89 ± 0.11	0.32 ± 0.01
BOD/COD	0.31 ± 0.01	0.48 ± 0.02

## Data Availability

Not applicable.

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
