# Peer review of "Bioelectrochemical Purification of Biomass Polymer Derived Furfural Wastewater and Its Electric Energy Recovery"

_polymers, 2023, doi:10.3390/polym15163422_

Round 1

Reviewer 1 Report

The paper investigates the removal of furfural from wastewater using microbial fuel cells. The study is designed to ensure that wastewater treatment is coupled with electricity generation. However, the novelty of the study could be questioned and the language is further hampering the expression of the scientific content.

Many studies have used MFC for the removal of furfural from wastewater; clearly indicate the novelty of this study.

It is not clear what was the external resistance of the system all along.

The authors must provide a SEM image of their anode electrode to see the buildup of microorganisms to support the interpretation of results.

The authors must provide a polarization curve to better explain the electrochemical properties of the MFC.

Provide a graph that clearly shows the dynamic of glucose and furfural removal overtime and the impact on current density.

Fig 5 and 6 do not add value given the inconsistent of voltage variation.

None of the figures or table show standard deviation although the authors mentioned that the experiments were done in triplicates.

The information in Figure 7 is repeated in Table 3.

The paper requires considerable language editing.

The referencing style in the references list should be consistent.

Need considerable language editing.

Author Response

We really appreciate you for your carefulness and conscientiousness. Your suggestions are really valuable and helpful for revising and improving our paper. According to your suggestions, we have made the following revisions on this manuscript:

  • The paper investigates the removal of furfural from wastewater using microbial fuel cells. The study is designed to ensure that wastewater treatment is coupled with electricity generation. However, the novelty of the study could be questioned and the language is further hampering the expression of the scientific content.Many studies have used MFC for the removal of furfural from wastewater; clearly indicate the novelty of this study.

Answer regarding to Comments:

MFC is of great concern in the field of waste water treatment and electricity generation. Recycling resources and energy from waste water has become a new trend of waste water treatment. Meanwhile, MFC shows a unique ability of degradation and electricity generation due to the difference of microbial population, anode substrate, reactor configuration, electrode materials and so on. Furfural is highly toxic to the respiratory system, nervous system, and blood system of the animals. Following the liver, spleen and kidney injury by furfural, there is progressive cancer. However, there is limited report on the application of MFC in furfural wastewater treatment. Luo et al. (2010) first reported that electricity was generated from the biodegradation of furfural in the MFC under the condition of low concentration furfural (the highest concentration is 6.68mM furfural, equivalent to 642mg/L). In our study, MFC is used to treat high-concentration furfural wastewater (up to 7000mg/L) and study the electricity generation performance. To our knowledge, this is the first study on Bioelectrochemical purification of high concentration furfural wastewater by MFC.

Luo, Y.; Liu, G.L.; Zhang, R.D.; Zhang, C.P. Power generation from furfural using the microbial fuel cell. J. Power Sources 2010195, 190-194.

  • The authors must provide a SEM image of their anode electrode to see the buildup of microorganisms to support the interpretation of results.

Answer regarding to Comments:

The surface morphology of clean carbon-brush, and biofilms attached on the carbon-brush surfaces in anode were observed by SEM photomicrographs (Fig. S1). The photomicrographs revealed that the carbon-brush in anode were regularly and compactly woven together, whereas those of MFC covered almost the entire surface area by a rich biofilm.

Fig. S1 SEM images of (A) clean carbon-brush, and (B) biofilms attached on the carbon-brush surfaces in anode.

  • The authors must provide a polarization curve to better explain the electrochemical properties of the MFC.

Answer regarding to Comments:

The maximum power density had reached 1280 mW/m2 when 1.0 g/L glucose was used as the substrate, but the power density was very low when furfural wastewater was used as the substrate (0.132mW/m2). I'm sorry for our incorrect description of the polarization curve when glucose is used as the substrate in the article. Figure 3 is the polarization curve when furfural wastewater is used as the substrate.

  • Provide a graph that clearly shows the dynamic of glucoseand furfural removal overtime and the impact on current density.

Answer regarding to Comments:

In the MFC system, glucose is first added as a substrate. When the voltage is 0, different concentrations of furfural are added successively. Therefore, it is not necessary to show the addition of glucose in the diagram. Fig. 5 and Fig. 6 effectively represent the dynamics of furfural removal over time and the effect of voltage. Due to the large gap between ordinates, in order to be more intuitive, I refer to the article (Wu et al. 2018) and do not merge Fig. 5 and Fig. 6.

Wu, H.; Fu, Y.; Guo, C.Y.; Li, Y.B.; Jiang, N.Z.; Yin, C.R. Electricity generation and removal performance of a microbial fuel cell using sulfonated poly (ether ether ketone) as proton exchange membrane to treat phenol/ acetone wastewater. Bioresource Technol. 2018, 260, 130-134.

  • Fig 5 and 6 do not add value given the inconsistent of voltage variation.

Answer regarding to Comments:

We uniformly calibrated the horizontal coordinate and resubmitted the pictures to the revised paper.

  • None of the figures or table show standard deviation although the authors mentioned that the experiments were done in triplicates.

Answer regarding to Comments:

Table 2 has removed the range and added standard deviation instead.

  • The information in Figure 7 is repeated in Table 3.

Answer regarding to Comments:

We deleted Table 3.

  • The paper requires considerable language editing.

Answer regarding to Comments:

We made language editing by MDPI after revising the paper.

  • The referencing style in the references list should be consistent.

Answer regarding to Comments:

We have carefully modified the format of the references as required.

Reviewer 2 Report

Hailing Tian et al, studied about the Bioelectrochemical Purification of Biomass Derived Furfural Wastewater and its Electric energy recoveryIt is a good approach but the paper can be accepted for publication only after major revision. It is recommended that the following aspects to be addressed by the authors.

Clearly state the objectives and scope of the present study in the introduction section.

Mention the harmful effects caused by the furfural wastewater in human and environment?

Which biomass polymer furfural wastewater was used in this study. Mention the name of biomass

What is the role of proton exchange membrane in MFC? Explain the principle behind it

What is the name of buffer which was added in cathode chamber and why it was added?

What was the initial pH present in furfural wastewater and how it was made neutral?

Where will be the attached growth of microbes happen in chambers?

 How does the treated wastewater is collected after treatment?

What about the anaerobic gas produced during treatment. How it was treated? Explain

Based on what criteria the microbes were selected for the study? Explain in detail about the culturing of selected microbes

Authors should provide the detailing about economic assessment of the study.

Conclusions should show clearly the results and impact of this study in wastewater treatment and electricity production. Mention the COD and BOD removal efficiency after the treatment

Future scope and limitations of this study should be discussed

In results and discussion section, the outcome of the present study should be compared with the existing literature to highlight the significance of the study.

Latest references need to be cited throughout the manuscript

Grammar check can be made throughout the manuscript

Grammar check can be made throughout the manuscript

Author Response

We really appreciate you for your carefulness and conscientiousness. Your suggestions are really valuable and helpful for revising and improving our paper. According to your suggestions, we have made the following revisions on this manuscript:

  • Clearly state the objectives and scope of the present study in the introduction section. Mention the harmful effects caused by the furfural wastewaterin human and environment?

Answer regarding to Comments:

We carefully revised the introduction as requested and mentioned the hazards of furfural wastewater to humans and the environment.

  • Which biomass polymer furfural wastewater was used in this Mention the name of biomass. What is the role of proton exchange membrane in MFC? Explain the principle behind it.

Answer regarding to Comments:

There are H2SO4, CH3COOH, hemicellulose, cellulose, lignin, furfural, total nitrogen and total phosphorus in furfural wastewater. Proton exchange membrane is used the solid proton exchange membrane in MFC to isolate oxygen and hydrogen, so that hydrogen oxidizes at the anode to produce electrons and protons, electrons flow into the cathode through the external current, protons pass through the proton exchange membrane to the cathode, and combine with oxygen to form water. The working principle of proton exchange membranes in MFC is based on a chemical reaction in which hydrogen gas is oxidized at the anode: H2→2H++2e-. Electrons flow through an external current into the cathode, creating an electric current, while protons penetrate through a solid polymer electrolyte, or proton exchange membrane. At the cathode, oxygen reacts with protons to form water: 1/2 O2+2H++2e-→H2O. The chemical equation of the whole reaction is H2+1/2 O2→H2O.

  • What is the name of buffer which was added in cathode chamber and why it was added?

Answer regarding to Comments:

The pre-prepared carbon-free buffer solution (PSB) was added to the cathode chamber. The composition of wastewater is complex, and the proton membrane with negative structure is seriously polluted. Adding buffer solution can increase the electricity generation rate.

  • What was the initial pH present in furfural wastewater and how it was made neutral?

Answer regarding to Comments:

The initial pH present in furfural wastewater was 2.13—2.17. In the MFC system, sulfuric acid is consumed in the hydrolysis of organic matter in wastewater. In addition, the acetic acid in furfural wastewater is easily degraded by microorganisms, thus increasing the pH value of furfural wastewater, reaching the standard of discharge (neutral).

  • Where will be the attached growth of microbeshappen in chambers?

Answer regarding to Comments:

The carbon-brush in anode were covered almost the entire surface area by the attached growth of microbes.

  • How does the treated wastewater is collected after treatment?

Answer regarding to Comments:

The treated wastewater has reached the discharge standard and can be discharged directly.

  • What about the anaerobic gas produced during treatment. How it was treated?

Answer regarding to Comments: 

The main biogas on MFC anode is carbon dioxide. In addition, MFC is currently in the research and development stage, not to achieve large-scale production and application, so have not considered waste gas treatment.

  • Based on what criteria the microbes were selected for the study? Explain in detail about the culturing of selected microbes.

Answer regarding to Comments:

The microorganisms used in this study were microbial consortia of activated sludge from sewage treatment plant, and no specific microorganisms were selected. 

9)Authors should provide the detailing about economic assessment of the study.

Answer regarding to Comments:

At present, MFC is still in the research and development stage, has not yet realized the large-scale production and the application, therefore has not yet reached the economic assessment stage. 

10) Conclusions should show clearly the results and impact of this study in wastewater treatment and electricity production. Mention the COD and BOD removal efficiency after the treatment.

Answer regarding to Comments:

As requested, the conclusion has been revised and mentioned the COD and BOD removal efficiency after the treatment.

11) Future scope and limitations of this study should be discussed In results and discussion section, the outcome of the present study should be compared with the existing literature to highlight the significance of the study. Latest references need to be cited throughout the manuscript Grammar check can be made throughout the manuscript. 

Answer regarding to Comments:

The discussion section has been revised as requested and the latest literature has been added to the references.

Reviewer 3 Report

1. It appears that Table 1 and Figure 3 contain overlapping information, so one of them should be excluded. Additionally, after conducting the three measurements and calculating the average value, it would be beneficial to include the standard deviation values as well. Furthermore, a detailed description of how long the external resistance was maintained after each change and before switching to another external resistance should be provided.

 2. Why is the voltage generation at the enrichment stage not presented when it reaches a steady-state? Moreover, it is common practice to measure the polarization curve under steady-state conditions. Is there a specific reason for measuring it in an unstable state?

3.  Although only furfural was analyzed, does it undergo direct reduction to inorganic carbon without generating harmful intermediate by-products during degradation? Can we be confident that there will be no issues when applied in an actual process?

4. The results of microbial cluster analysis are insufficient, and the purpose of cluster interpretation is unclear. By observing the changes in clusters before and after the introduction of the initial strains and furfural wastewater, important microbial information can be identified, enabling further investigation. Merely presenting the proportion of microbial clusters would not be sufficient to draw meaningful conclusions.

 Although only furfural was analyzed, does it undergo direct reduction to inorganic carbon without generating harmful intermediate by-products during degradation? Can we be confident that there will be no issues when applied in an actual process?

Author Response

We really appreciate you for your carefulness and conscientiousness. Your suggestions are really valuable and helpful for revising and improving our paper. According to your suggestions, we have made the following revisions on this manuscript: 

  • It appears that Table 1 and Figure 3 contain overlapping information, so one of them should be excluded. Additionally, after conducting the three measurements and calculating the average value, it would be beneficial to include the standard deviation values as well. Furthermore, a detailed description of how long the external resistance was maintained after each change and before switching to another external resistance should be provided.

Answer regarding to Comments:

We removed Table 1 as requested and reflected the standard variance in Table 2.

  • Why is the voltage generation at the enrichment stage not presented when it reaches a steady-state? Moreover, it is common practice to measure the polarization curve under steady-state conditions. Is there a specific reason for measuring it in an unstable state?

Answer regarding to Comments:

Our constructed MFC was directly treated for furfural wastewater after reaching a maximum voltage after 700 hours of operation, so voltage stability is not shown in Figure 2. Figure 3 shows the power density and polarization curve during the treatment of furfural wastewater, which is the polarization curve and power density curve of MFC after adding high concentration furfural wastewater.

3) Although only furfural was analyzed, does it undergo direct reduction to inorganic carbon without generating harmful intermediate by-products during degradation? Can we be confident that there will be no issues when applied in an actual process?

Answer regarding to Comments:

The degradation pathways of furfural have been extensively reported by Zamt et al (J. Bacteriol. 2001, 183, 1954-1960), Zamt et al (Chin. Environ. Sci. 200020, 241-244), and Brune et al (Appl. Environ. Microbiol. 1983, 1, 1187-1192.).

Under aerobic conditions:

Under anaerobic conditions:

In our system, the degradation of furfural is carried out under anaerobic conditions. The product is mainly used to supply carbon sources to electric-producing microorganisms, so it can be determined that all carbon sources are degraded into carbon dioxide. Therefore, it is very safe to degrade furfural with MFC, and the end product is only carbon dioxide and water.

4) The results of microbial cluster analysis are insufficient, and the purpose of cluster interpretation is unclear. By observing the changes in clusters before and after the introduction of the initial strains and furfural wastewater, important microbial information can be identified, enabling further investigation. Merely presenting the proportion of microbial clusters would not be sufficient to draw meaningful conclusions.

 Answer regarding to Comments:

We are sorry that the microbial cluster heat map was not provided to us by Sangon Biotech (Shanghai) Co., Ltd. The original data cannot be found. Therefore, the microbial community structure, species taxonomic abundance and phylogenetic tree map are provided with supplementary data.

Supporing Information

Fig.S2 Community structure distribution map of samples at Genus level

Fig.S3 Taxonomic abundance histogram of samples at Genus level

Round 2

Reviewer 1 Report

Approved with consideration of final editing.

Final editing needed

Author Response

We really appreciate you for your carefulness and conscientiousness. Your suggestions are really valuable and helpful for revising and improving our paper.  According to your request, we re-generate the PDF and upload the attachment.

Reviewer 2 Report

No comments

Author Response

Dear reviewer:

Thank you for your decision on my manuscript.

Reviewer 3 Report

Manuscript has been significantly improved. However, there are some parts that appear to require further revisions. 

1. In Fig 4, it is common to represent values on the y-axis as exponentials when they are large. 

2. Detailed explanations for the symbols and texts in Figs should also be included in the figure caption.  For ex, arrow in Fig 5 , and A, B, ~~, and F in Fig 6.

3. There is a lack of detailed information about how the analysis was conducted. Instead of simply stating that external resistance was changed. Pleased provide a more specific description of the methods employed. 

4. Despite the fact that approximately 1.3 g/L of BOD concentration was removed at an influent concentration of around 5 g/L, and the current efficiency (CE) exceeded 91%, the extremely low level of electricity generation raises the question of why this is the case. Generally, when using glucose or acetate as substrates, the CE typically ranges from 60% to 70%. In real wastewater scenarios, CE can even drop to below 10%. Given this context, it is essential to identify the factors responsible for the significantly higher CE observed in this experiment.

Author Response

Dear reviewer:

Thank you for your decision and constructive comments on my manuscript. We have carefully considered the suggestion of Reviewer and make some changes. We have tried our best to improve and made some changes in the manuscript.

The yellow part that has been revised according to your comments. Revision notes, point-to-point, are given as follows:

  1. In Fig 4, it is common to represent values on the y-axis as exponentials when they are large.

Answer regarding to Comments:

We redrew Figure 4 as requested and submitted it to the revised paper.

  1. Detailed explanations for the symbols and texts in Figs should also be included in the figure caption.  For ex, arrow in Fig 5 , and A, B, ~~, and F in Fig 6.2.

Answer regarding to Comments:

In Figures 5 and 6, the arrows represent the time of adding furfural wastewater during MFC operation, A-F representing the voltage and removal rate under different concentrations of furfural wastewater. The furfural concentrations of A-F were supplemented in Figure 5 and 6 title of revised manuscript.

  1. There is a lack of detailed information about how the analysis was conducted. Instead of simply stating that external resistance was changed. Pleased provide a more specific description of the methods employed.

Answer regarding to Comments:

The power density (PD) and polarization curve of the MFC were plotted by changing the external resistance (50–9999Ω). The electrochemical performance of the MFC was evaluated through the expression V = IR (Ohm's Law), where V is voltage (V), I is current (A), and R is external resistance (Ω). Power (P) in watts (W) was calculated from P = IV. The power density was calculated according to the electrodes' total surface area. The internal resistance was determined from the slope of the polarization curve. The relevant contents are marked in yellow in the revised paper.

  1. Despite the fact that approximately 1.3 g/L of BOD concentration was removed at an influent concentration of around 5 g/L, and the current efficiency (CE) exceeded 91%, the extremely low level of electricity generation raises the question of why this is the case. Generally, when using glucose or acetate as substrates, the CE typically ranges from 60% to 70%. In real wastewater scenarios, CE can even drop to below 10%. Given this context, it is essential to identify the factors responsible for the significantly higher CE observed in this experiment.

Answer regarding to Comments:

The 91% mentioned in the article should be the degradation efficiency of furfural reached 91% (furfural concentration at 7000 mg/L). In the conclusion, the statement of "coulombic efficiency of about 91%" is incorrect, and has been revised in the manuscript. I'm very sorry!